# Near-Field Enhancement and Polarization Selection of a Nano-System for He-Ne Laser Application

**DOI:** 10.3390/nano9101421

**Published:** 2019-10-06

**Authors:** Qiao Wang, Shuwen Chu, Li Yu, Huixuan Gao, Wei Peng

**Affiliations:** Department of Physics, Dalian University of Technology, Ganjingzi District, Dalian 116024, China; wangqiao@dlut.edu.cn (Q.W.); G21402062@mail.dlut.edu.cn (S.C.); 1050027560@mail.dlut.edu.cn (L.Y.); shark@mail.dlut.edu.cn (H.G.)

**Keywords:** nano-system, extinction ratio, transmission, FDTD method

## Abstract

In this paper, we focus on transmission behavior based on the single aperture with a scatter. Both the near-field enhancement and polarization selection can be achieved numerically with a proposed nano-system under He-Ne laser wavelength. The nano-system consists of an Ag antenna, a wafer layer, an Ag film with an aperture and a dielectric substrate. Numerical results show that the near-field enhancement is related to the FP-like resonance base on surface plasmon polaritons (SPPs) in the metal–isolator–metal (MIM) waveguide for transverse magnetic (TM) polarization. The near-field optical spot is confined at the aperture export with a maximal electric intensity 20 times the value of the incident field for an antenna length of 430 nm. The transmission cutoff phenomenon for transverse electric (TE) polarization is because the transmission is forbidden for smaller aperture width. High extinction ratios of $9.6\times 10^{-8}$ (or 70.2 dB) and $4.4\times 10^{-8}$ (or 73.6 dB) with antenna lengths of 130 nm and 430 nm are achieved numerically with the nano-system. The polarization selective property has a good angular tolerance for oblique angles smaller than 15°. The spectral response is also investigated. We further demonstrate that the nano-system is applicable for another incident wavelength of 500 nm. Our investigation may be beneficial for the detection of polar molecules or local nano polarized nanosource.

## 1. Introduction

The extraordinary optical transmission (EOT) has aroused great attention since it was first observed in the periodic aperture arrays in a metallic film by T. W. Ebbesen et al. in 1998 [1]. Many subsequent works have been conducted numerically and experimentally with different materials, film thickness, and aperture shapes, in order to understand the fundamental physics involved [2,3,4,5,6]. It is generally accepted that the surface plasmon polaritons (SPPs) play an important role in the EOT phenomenon with periodic aperture arrays. The SPPs are light coupled to collective oscillations of free electrons at a noble metal/dielectric interface. The unique surface nature of SPPs yields a wide range of applications, such as nanolithography [7,8], biosensing [9,10,11,12], enhanced spectroscopy [13,14,15,16], as well as solar cell [17,18,19,20].

Besides the EOT with periodic aperture arrays, the enhanced transmission through a single aperture is an important issue as it can achieve transmission enhancement in the optical near-field, which is useful for applications of nanofocusing, nanolithography and nanoimaging. A. Degiron et al. and F. J. Garcia-Vidal et al. investigated the EOT behavior through a single circular aperture [21] and a single rectangular aperture [22], respectively. Subsequently, ridge aperture with different shapes, such as C-shape, H-shape, and bowtie-shape, have aroused great interest as a key element for near-field optical enhancement. X. Shi et al. reported λ/10 resolution and resonant transmission with C-shaped aperture [23]. N. C. Lindquist et al. demonstrated a plasmonic nanofocusing with a metallic pyramid and an integrated C-shaped aperture [24]. X. Xu’s group studied the SPPs enhanced EOT of a C-shaped and an H-shaped nanoaperture in a silver film and the physical mechanism of the EOT was attributed to the waveguide mode and Fabry–Perot-like resonance [25]. Then, they experimentally demonstrated an enhanced optical near-field of a bowtie aperture in an aluminum film using near-field scanning optical microscopy [26]. They further experimentally performed a 66 nm linewidth for nanoimaging [27] and a 24 nm linewidth for nanolithography [28] on a positive photoresist coated sample, using near-field transmission enhancement with a bowtie aperture integrated on an NSOM probe.

Recently, another method of near-field transmission enhancement with a single aperture was found by introducing a scatter or meta-structure near the neighbourhood of the aperture. S. He’s group reported an EOT through a vertical nanoslit by introducing an antenna over the slit opening [29]. P. Banzer et al. experimentally demonstrated an enhanced transmission through a single coaxial aperture when compared to a circular aperture with the same outer diameter [30]. F. J. Valdivia-Valero et al. studied the transmission enhancement of a subwavelength 2D slit by exciting the eigenmodes of plasmonic cylinders at the aperture entrance [31]. K. Bi’s group reported an enhanced transmission by utilizing the Mie resonances of two high-permittivity low-loss ceramic particles located at either side of a metallic aperture [32]. Then, they experimentally and theoretically demonstrated the magnetically tunable dual-band transmission by placing two pairs of dielectric cubes and ferrite cuboids symmetrically on both sides of the single subwavelength aperture [33,34]. Thermally tunable enhanced transmission through a subwavelength metallic aperture was also realized by a dielectric-based metamaterial resonator with high temperature coefficient [35]. An enhanced transmission through a single subwavelength aperture incorporated with meta-structure was also reported by different research groups [36,37,38,39,40]. Afterwards, an enhanced electromagnetic transmission through a metallic aperture was realized with a dielectric meta-atom placed into the aperture at the meta-atom’s resonant frequency [41]. All these works investigate the near-field enhanced transmission with a single aperture.

In this paper, we also focus on the transmission behavior based on the single aperture with an antenna. Both the near-field enhancement and polarization selection can be achieved numerically with the proposed nano-system under He-Ne laser wavelength. The nano-system consists of an Ag antenna, a wafer layer, an Ag film with an aperture and a dielectric substrate. Numerical results show that the near-field enhancement is related to the FP-like resonance base on surface plasmon polaritons (SPPs) in the metal–isolator–metal (MIM) waveguide for transverse magnetic (TM) polarization. The near-field optical spot is confined at the aperture export with maximal electric intensity 20 times the value of the incident field for an antenna length of $l_{a}=430\;\mathrm{nm}$. The transmission cutoff phenomenon for transverse electric (TE) polarization is because the transmission is forbidden for smaller aperture width $w_{b}$. High extinction ratios of $9.6\times 10^{-8}$ (or 70.2 dB) and $4.4\times 10^{-8}$ (or 73.6 dB) with antenna lengths of $l_{a}=130\;\mathrm{nm}$ and $l_{a}=430\;\mathrm{nm}$ are achieved numerically with the nano-system. The polarization selective property of the nano-system has a good angular tolerance for oblique angles smaller than $15^{{}^{\circ}}$. The spectral response of the nano-system is also investigated. We demonstrate that the design of the nano-system is applicable for another incident wavelength of 500 nm. Our investigation may be beneficial for the detection of polar molecules or localized nano polarized source.

## 2. Structures and Methods

The proposed nano-system is composed of four parts: a silver (Ag) nano-antenna, a SiO_2_ wafer layer, a silver (Ag) film with an aperture and a sustained SiO_2_ substrate. The top and side views of the nano-system are shown in Figure 1a,b. The nano-antenna and the aperture are designed to be perpendicular to each other for different polarizations. The antenna is defined by its length of $l_{a}$ and width of $w_{a}$. The aperture is described by its length of $l_{b}$ and width of $w_{b}$. The thicknesses of the nano-antenna, the wafer layer and the silver film are represented as $t_{a}$, $t_{g}$ and $t_{b}$, respectively. The electromagnetic wave injects from the top side with its wavenumber k and angle θ with respect to the *z* axis. The electric field component parallel to the *x* or *y* axis indicates a TM or TE polarization.

The complex permittivity of the noble metal (silver in this paper) depends on the incident frequency and can be described by the Drude model as
(1)$$\varepsilon \left(\omega \right)=\varepsilon_{\infty }-\frac{\omega_{p}^{2}}{\omega^{2}+i\omega \gamma_{0}},$$
where ω is the angular frequency of incident wave, $\varepsilon_{\infty }$ represents the relative permittivity at infinite frequency, $\omega_{p}$ refers to the bulk plasma frequency and γ stands for the electron collision rate. In our simulation, these parameters are $\varepsilon_{\infty }=5$, $\omega_{p}=9.5\;\mathrm{eV}$ and $\gamma_{0}=0.015\;\mathrm{eV}$, respectively. The refractive index of SiO_2_ material is a constant of 1.45.

We used a finite-difference time-domain (FDTD) method for the investigation in this paper. The basic idea of the FDTD method is to use the central difference instead of the partial derivative of the field component in Maxwell’s equations to simulate the wave propagation process in the time domain. The method was first proposed by K. S. Yee in 1966 [42]. He divided the computational space into individual cubes, called the Yee cell. In the Yee cell, the sampling nodes of electromagnetic field components are arranged alternately in time and space. Each electric (magnetic) field component is surrounded by four magnetic (electric) field components, which satisfies the requirement of Faraday’s law and Ampere circuital theorem. The field distribution is finally obtained by solving the field component in Maxwell’s equation step by step in time axis. The simulation space is 1200 nm×1200 nm×1400 nm and the mesh size is 5 nm×5 nm×5 nm. The perfectly matched layers (PML) are applied on all the boundaries to fully absorb the outgoing wave of the simulation space.

## 3. Results and Discussions

We start with a fixed wavelength of 632.8 nm under normal incidence, as the He-Ne laser is the most common laser applied in the visible region. The complex permittivity of silver is −18.35+0.48i under the wavelength. The parameters of the aperture are constant as $l_{b}=200\;\mathrm{nm}$ and $w_{b}=30\;\mathrm{nm}$. The thicknesses of the nano-antenna, the wafer layer and the silver film are $t_{a}=50\;\mathrm{nm}$, $t_{g}=50\;\mathrm{nm}$ and $t_{b}=220\;\mathrm{nm}$. The electric intensity of incident field is 1V/m. The width of the nano-antenna is $w_{a}=50\;\mathrm{nm}$. The influence of the antenna length of $l_{a}$ on transmission is investigated under TM-polarized light. Figure 2a shows the absolute transmission with various antenna lengths of $l_{a}$ from 30 nm to 600 nm, every 20 nm. As the antenna length changes, two peaks with antenna lengths $l_{a}$ of 130 nm and 430 nm appear in the transmission in Figure 2a.

Figure 2b,c show detailed electric and magnetic field distributions of $l_{a}=130\;\mathrm{nm}$ and $l_{a}=430\;\mathrm{nm}$ for TM polarization in the *xz* section. From these two figures, we can see that the Ag antenna, wafer layer and Ag film form a horizontal MIM waveguide and the aperture can be taken as a disturbance of the waveguide. In the waveguide, the dispersion relationship of SPPs can be derived from Maxwell’s equations and boundary conditions as [43]
(2)$$e^{-2k_{2}t_{g}}=\frac{\frac{k_{2}}{\varepsilon_{2}}+\frac{k_{1}}{\varepsilon_{1}}}{\frac{k_{2}}{\varepsilon_{2}}-\frac{k_{1}}{\varepsilon_{1}}}\cdot \frac{\frac{k_{2}}{\varepsilon_{2}}+\frac{k_{3}}{\varepsilon_{3}}}{\frac{k_{2}}{\varepsilon_{2}}-\frac{k_{3}}{\varepsilon_{3}}},$$
where the quantities with subscripts of 1, 2 and 3 represent the corresponding wavenumber and permittivity of the Ag antenna, wafer layer and Ag film, respectively. An FP-like resonance base on SPPs appears in the MIM waveguide. The FP-like resonance is related to the slit thickness in an open FP cavity [44]. The condition of the FP-like resonance is written as [45]
(3)$$\frac{4\pi l_{a}}{\lambda_{\mathrm{spp}}}+\arg\left(\rho_{1}\rho_{2}\right)=2m\pi .$$
where $\lambda_{\mathrm{spp}}$ represents the SPPs wavelength in the MIM waveguide, m is an integer, $\rho_{1}$ and $\rho_{2}$ denote the reflective coefficients at the two ends of the nano-antenna, respectively. According to the equation, $l_{a}=130\;\mathrm{nm}$ and $l_{a}=430\;\mathrm{nm}$ correspond to the FP-like resonance with m=0 and m=1. When the antenna length $l_{a}$ reaches the FP-like resonant condition, the transmission is maximal, i.e., 0.09 with $l_{a}=130\;\mathrm{nm}$ and 0.21 with $l_{a}=430\;\mathrm{nm}$, because the antenna helps the energy focus on the aperture entrance and then transmit through the aperture. Figure 2d,e show the distribution of the spots at the aperture export for TM polarization in the *xy* section, which indicates that the electric field is localized at the near-field of the aperture export for both antenna lengths. The maximum electric intensity of these spots is 12 times and 20 times the value of the incident field for $l_{a}=130\;\mathrm{nm}$ and $l_{a}=430\;\mathrm{nm}$, respectively.

Subsequently, the influence of polarization angle is investigated. The antenna length is chosen as $l_{a}=130\;\mathrm{nm}$ and $l_{a}=430\;\mathrm{nm}$ from Figure 2a. Figure 3a shows the transmission with polarization angles from $0^{{}^{\circ}}$ to $360^{{}^{\circ}}$, every $5^{{}^{\circ}}$. The polarization angle of $0^{{}^{\circ}}$ and $180^{{}^{\circ}}$ corresponds to the TM polarization, while the polarization angle of $90^{{}^{\circ}}$ and $270^{{}^{\circ}}$ stands for the TE polarization. For TM polarization, the transmission is 0.09 for $l_{a}=130\;\mathrm{nm}$ and 0.21 for $l_{a}=430\;\mathrm{nm}$, as discussed in Figure 2a. For TE polarization, the transmission is $8.83\times 10^{-9}$ for $l_{a}=130\;\mathrm{nm}$ and $9.16\times 10^{-9}$ for $l_{a}=430\;\mathrm{nm}$. Figure 3b,c show the detailed electric and magnetic field distributions of $l_{a}=130\;\mathrm{nm}$ and $l_{a}=430\;\mathrm{nm}$ for TE polarization in the *xz* section. The two figures reveal that the energy is localized in the MIM waveguide and does not transmit through the aperture for both the antenna lengths. Figure 3d,e further give out the distributions at the aperture export for TE polarization in the *xy* section. This shows that there is no near-field spot at the aperture export for TE polarization with both antenna lengths. Note that the range of the color bar is from 0 to 1 in Figure 3d,e.

To quantitatively describe the polarization selective characteristic of the system, we introduce the concept of extinction ratio γ as
(4)$$\gamma =T_{\mathrm{TE}}/T_{\mathrm{TM}},$$
or
(5)$$\gamma =-10\times \mathrm{lg}(T_{\mathrm{TE}}/T_{\mathrm{TM}})\;\left(\mathrm{dB}\right),$$
where $T_{\mathrm{TE}}$ and $T_{\mathrm{TM}}$ are the transmissions for the TE and TM polarizations. The extinction ratio of γ is $9.6\times 10^{-8}$ (or 70.2 dB) for $l_{a}=130\;\mathrm{nm}$ and $4.4\times 10^{-8}$ (or 73.6 dB) for $l_{a}=430\;\mathrm{nm}$ according to Equation (4). A system with a high extinction ratio can be realized for the both antenna lengths.

To reveal the physics mechanism of the proposed perfect system, we further examine the transmission with different aperture parameters. Figure 4a,b show the transmission with different aperture widths and lengths for the TE and TM polarizations. The antenna is defined as $l_{a}=430\;\mathrm{nm}$ and $w_{a}=50\;\mathrm{nm}$. The thicknesses of the nano-antenna, the wafer layer and the silver film are $t_{a}=50\;\mathrm{nm}$, $t_{g}=50\;\mathrm{nm}$ and $t_{b}=220\;\mathrm{nm}$. The aperture length remains at $l_{b}=200\;\mathrm{nm}$ as the aperture width $w_{b}$ changes in Figure 4a, while aperture width is constant at $w_{b}=30\;\mathrm{nm}$ with different aperture lengths $l_{b}$ in Figure 4b. Figure 4a shows that the aperture width $w_{b}$ plays a critical role in the transmission cutoff phenomenon for TE polarization of the nano-system. When $w_{b}\leq 100\;\mathrm{nm}$, the transmission for TE polarization is smaller than 0.03. This is because the light transmission with an electric field parallel to the *y* axis is forbidden for smaller $w_{b}$. A similar transmission cutoff at half wavelength has been reported for a nano-slit with an electric field parallel to the slit in reference [46]. The difference between the critical values of the transmission cutoff in our system and the reference is probably due to the structural difference between the two works. In our system, it is a rectangle aperture, while only a grating slit is presented in the reference. When $w_{b}$ exceeds 100 nm, the transmission cutoff disappears for TE polarization and the nano-system is not suitable for a nano-system. The transmission for TM polarization decreases dramatically as $w_{b}$ increases. Therefore, one should keep the width of the aperture below 100 nm when designing such a nano-system. As Figure 4b illustrates, the transmission remains below $1.2\times 10^{-8}$ for TE polarization, as $l_{b}$ changes from 30 nm to 600 nm. The transmission for TM polarization peaks at 200 nm and the full width at half maximum (FWHM) is about 80 nm. Consequently, the $l_{b}$ should be chosen from 160 nm to 240 nm, and the best value of $l_{b}$ is 200 nm for the nano-system.

Figure 4c,d show the corresponding extinction ratios with different $w_{b}$ and $l_{b}$. The extinction ratio is smaller than 1% with $w_{b}\leq 100\;\mathrm{nm}$. The transmissions for the TM and TE polarizations are comparable with large $w_{b}$, thus the extinction ratio increases. The extinction ratio drops dramatically as $l_{b}$ increases because transmissions for the TM and TE polarizations are comparable with small $l_{b}$ and transmission is cut off for the TE polarization with large $l_{b}$.

To make explicit that the transmission cutoff is not related to the nano-antenna, we also compare the transmission of different $w_{b}$ and $l_{b}$ with and without antenna for the TE polarization in Figure 5. Figure 5a shows that with or without antenna, the transmission curves with $w_{b}$ have a similar trend, i.e., transmission cutoff remains with $w_{b}\leq 100\;\mathrm{nm}$ and disappears with $w_{b}>100\;\mathrm{nm}$. Figure 5b illustrates the same transmission trend for the TE polarization as $l_{b}$ changes. The results exclude the influence of nano-antenna and support our explanation that the aperture width $w_{b}$ is the critical reason for the generation of the transmission cutoff for the TE polarization on another aspect.

Since all previous discussions are based on normal incidence, we then study the nano-system with oblique incidence for practical applications. The antenna width is defined as $w_{a}=50\;\mathrm{nm}$. The aperture size is $l_{b}=200\;\mathrm{nm}$ and $w_{b}=30\;\mathrm{nm}$. The thicknesses of the nano-antenna, the wafer layer and the silver film are $t_{a}=50\;\mathrm{nm}$, $t_{g}=50\;\mathrm{nm}$ and $t_{b}=220\;\mathrm{nm}$. Figure 6a clearly illustrates the gradual decrease of transmission as θ increases with $l_{a}=130\;\mathrm{nm}$ and $l_{a}=430\;\mathrm{nm}$ for TM polarization. The magnitude of transmission is on the order of $10^{-9}$ or $10^{-8}$ with θ from $0^{{}^{\circ}}$ to $30^{{}^{\circ}}$ at $l_{a}=130\;\mathrm{nm}$ and $l_{a}=430\;\mathrm{nm}$ for the TE polarization. The transmission is larger than its half maximum for $\theta <15^{{}^{\circ}}$, indicating that the proposed nano-system is suitable for oblique angles smaller than 15°. Figure 6b shows the transmission with different polarization angles for oblique angles θ from $0^{{}^{\circ}}$ to $30^{{}^{\circ}}$, every $5^{{}^{\circ}}$. The larger the θ is, the smaller transmission received.

To reveal the gradual decrease of transmissions for TM polarization as θ increases, we illustrate the electric and magnetic field distributions with $\theta =10^{{}^{\circ}},\;20^{{}^{\circ}}$ and $30^{{}^{\circ}}$ for the TM polarization in Figure 7. This shows that the energy localizations shift to the left side of the aperture entrance with an increasing oblique angle of θ. As a result, the transmission decreases gradually.

The spectral responses of the nano-system with $l_{a}=130\;\mathrm{nm}$ and $l_{a}=430\;\mathrm{nm}$ for the TM and TE polarizations are shown in Figure 8a. The magnitude of transmission is on the order from $10^{-10}$ to $10^{-7}$ for both antenna lengths for TE polarization, while there are three peaks in spectrum for both antenna lengths for the TM polarization. For the TM polarization with $l_{a}=430\;\mathrm{nm}$, the second and third peaks of the transmission are not obvious, so the spectral FWHM of the nano-system is mainly determined by the first peak and is about 40 nm. For the TM polarization with $l_{a}=130\;\mathrm{nm}$, the three peaks are shown clearly and the spectral FWHM contains two bands. The first narrow band is from 620 nm to 660 nm, and the second, wide band is from 730 nm to 970 nm. Then, we focus on the physical mechanism of the three peaks with $l_{a}=130\;\mathrm{nm}$ because they are more obvious. The first peak is related to FP-like resonance with m=0 in the MIM waveguide, as shown in Figure 2b. Figure 8b,c show the electric and magnetic field distributions of the second peak, with λ=790 nm and third peak, with λ=936 nm. Comparing the field distribution in Figure 2b and Figure 3b, it is found that the second peak is related to the resonance of the aperture. At the third peak, the nano-antenna can be viewed as a particle that concentrates light energy at the aperture entrance to help transmission.

Finally, we have tested the wavelength of 500 nm to identify whether the proposed nano-system is suitable for another wavelength. Figure 9a shows the transmission with various antenna lengths of $l_{a}$ from 30 nm to 600 nm for the TM polarization. The parameters of the aperture are $l_{b}=200\;\mathrm{nm}$ and $w_{b}=30\;\mathrm{nm}$. The thicknesses of the nano-antenna, the wafer layer and the silver film are $t_{a}=50\;\mathrm{nm}$, $t_{g}=50\;\mathrm{nm}$ and $t_{b}=200\;\mathrm{nm}$. The width of the nano-antenna is $w_{a}=50\;\mathrm{nm}$. The antenna length $l_{a}=230\;\mathrm{nm}$ corresponds to the FP-like resonance m=1 in the MIM waveguide for λ=500 nm. With the fixed antenna length, the transmission with various polarization angles from $0^{{}^{\circ}}$ to $360^{{}^{\circ}}$, every $5^{{}^{\circ}}$ is shown in Figure 9b. This shows that the polarization-selective phenomenon can also be realized for λ=500 nm with a high extinction ratio of $9.8\times 10^{-7}$ (or 60.1 dB), which demonstrates the proposed nano-system for another wavelength.

## 4. Conclusions

In summary, we propose a nano-system consisting of an Ag antenna, a wafer layer, an Ag film with an aperture and a dielectric substrate. Numerical results show that both near-field enhancement and polarization selection can be achieved with the nano-system under He-Ne laser wavelength. The near-field enhancement is related to the FP-like resonance base on surface plasmon polaritons (SPPs) in the metal–isolator–metal (MIM) waveguide for TM polarization. The near-field optical spots are confined at the aperture exports with maximal electric intensities 12 times and 20 times the value of the incident field for antenna lengths of $l_{a}=130\;\mathrm{nm}$ and $l_{a}=430\;\mathrm{nm}$. The transmission cutoff phenomenon for TE polarization is because the transmission is forbidden for smaller aperture width $w_{b}$. High extinction ratios of $9.6\times 10^{-8}$ (or 70.2 dB) and $4.4\times 10^{-8}$ (or 73.6 dB) with antenna lengths of $l_{a}=130\;\mathrm{nm}$ and $l_{a}=430\;\mathrm{nm}$ are achieved numerically with the nano-system. The nano-system has a good angular tolerance for oblique angles smaller than $15^{{}^{\circ}}$, although the transmission for the TM polarization gradually decreases as oblique angle θ increases. The decrease in transmission results from the shifting of energy localization with an increasing θ. The spectral FWHM of the nano-system with $l_{a}=430\;\mathrm{nm}$ is about 40 nm, while it contains two bands of 620–660 nm and 730–970 nm with $l_{a}=130\;\mathrm{nm}$. The design of the nano-system is also applicable to another incident wavelength of 500 nm.

In addition, the proposed nano-system may be fabricated using modern processing technologies, including focused ion beam (FIB) etching, vacuum evaporation, plasma-enhanced chemical vapour deposition (PECVD), wet etching, joining and adhesion technology. The possible preparation process is explained as follows: firstly, an Ag film is deposited on a polished SiO_2_ substrate by the vacuum evaporation method. Then a rectangular hole is fabricated using FIB. Simultaneously, a SiO_2_ film is deposited on a polished silicon substrate through PECVD. Subsequently, an Ag film is evaporated on the SiO_2_ film, and then an Ag antenna is fabricated with FIB. Afterwards, the silicon substrate is removed by wet etching. Finally, the two structures are bonded with thermal adhesives. Close attention should be paid to the amount of adhesive to ensure that the aperture is not blocked.

## Figures and Tables

**Figure 1 nanomaterials-09-01421-f001:**
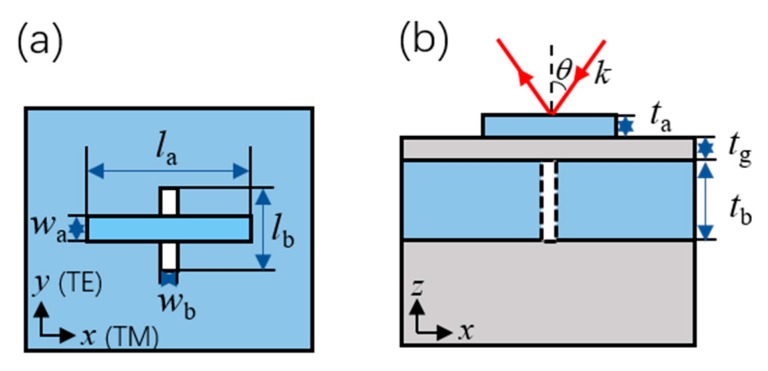
Schematic illustration of the high extinction ratio nano-system: (**a**) top view and (**b**) side view.

**Figure 2 nanomaterials-09-01421-f002:**
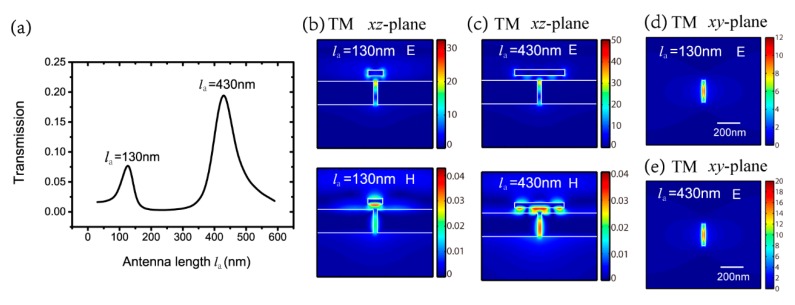
(**a**) Transmission with various antenna lengths of $l_{a}$ for transverse magnetic (TM) polarization; electric and magnetic field distributions of (**b**) $l_{a}=130\;\mathrm{nm}$ and (**c**) $l_{a}=430\;\mathrm{nm}$ for TM polarization in the *xz* section; electric field distributions of (**d**) $l_{a}=130\;\mathrm{nm}$ and (**e**) $l_{a}=430\;\mathrm{nm}$ for TM polarization at the aperture export.

**Figure 3 nanomaterials-09-01421-f003:**
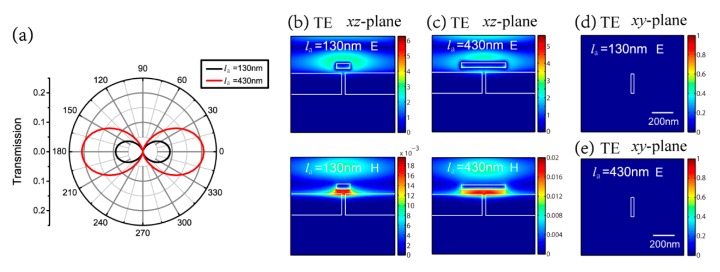
(**a**) Transmission with different polarization angles at $l_{a}=130\;\mathrm{nm}$ and $l_{a}=430\;\mathrm{nm}$; electric and magnetic field distributions of (**b**) $l_{a}=130\;\mathrm{nm}$ and (**c**) $l_{a}=430\;\mathrm{nm}$ for transverse electric (TE) polarization in the *xz* section; electric field distributions of (**d**) $l_{a}=130\;\mathrm{nm}$ and (**e**) $l_{a}=430\;\mathrm{nm}$ for TE polarization at the aperture export.

**Figure 4 nanomaterials-09-01421-f004:**
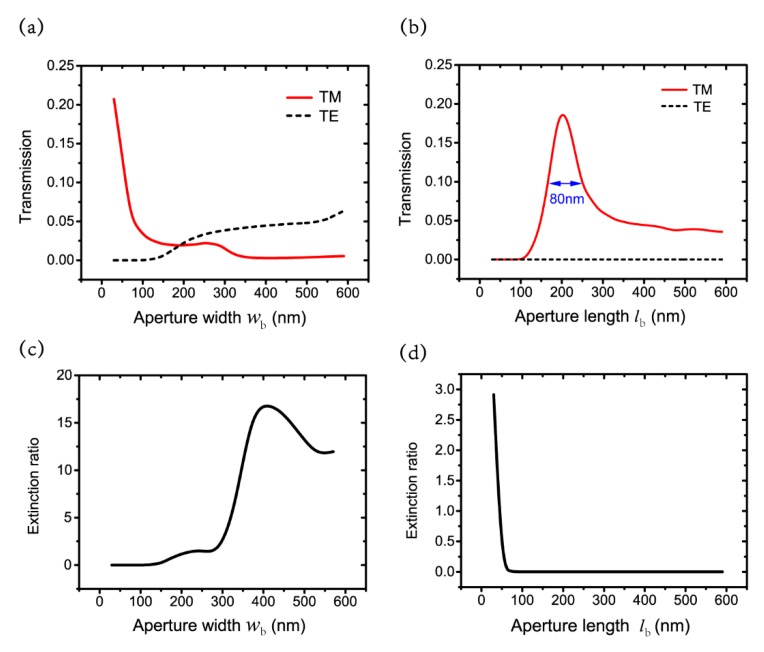
Transmissions with different (**a**) $w_{b}$ and (**b**) $l_{b}$ of the aperture for TM and TE polarizations; extinction ratio with different (**c**) $w_{b}$ and (**d**) $l_{b}$.

**Figure 5 nanomaterials-09-01421-f005:**
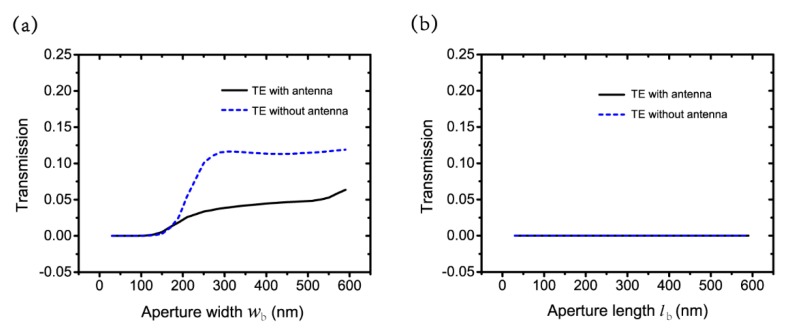
Transmission of different (**a**) $w_{b}$ and (**b**) $l_{b}$ with and without antenna for TE polarization.

**Figure 6 nanomaterials-09-01421-f006:**
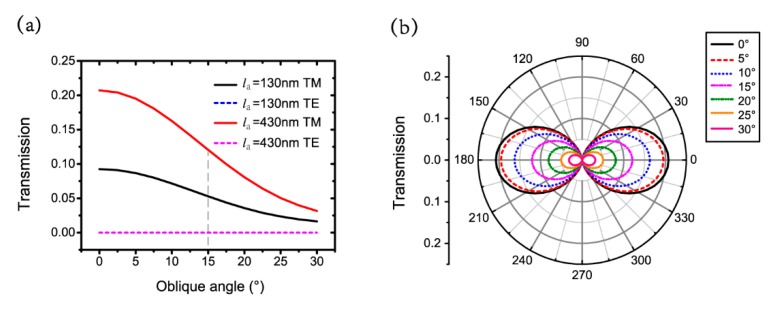
(**a**) Transmissions with different θ at $l_{a}=130\;\mathrm{nm}$ and $l_{a}=430\;\mathrm{nm}$ for TM and TE polarizations; (**b**) transmission with different polarization angles for different θ at $l_{a}=430\;\mathrm{nm}$.

**Figure 7 nanomaterials-09-01421-f007:**
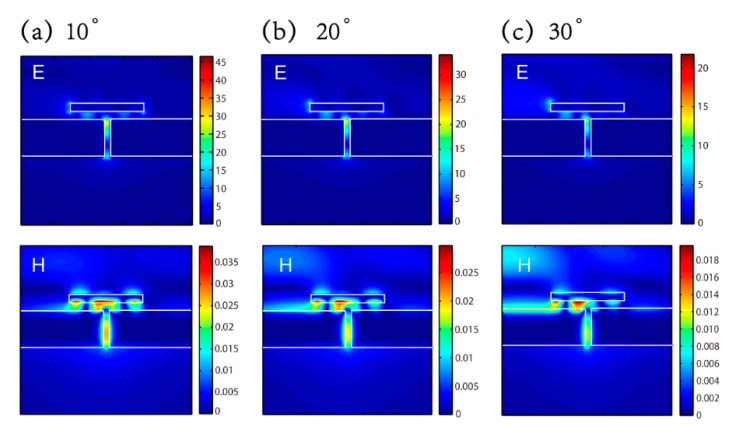
Electric and magnetic field distributions with θ of (**a**) $10^{{}^{\circ}}$, (**b**) $20^{{}^{\circ}}$ and (**c**) $30^{{}^{\circ}}$ for TM polarization.

**Figure 8 nanomaterials-09-01421-f008:**
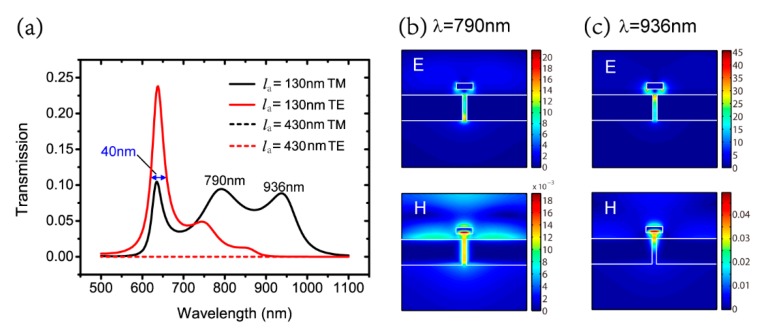
(**a**) Spectral responses of the nano-system with $l_{a}=130\;\mathrm{nm}$ and $l_{a}=430\;\mathrm{nm}$ for TM and TE polarizations; electric and magnetic field distributions of (**b**) λ=790 nm and (**c**) λ=936 nm with $l_{a}=130\;\mathrm{nm}$ for TM polarization.

**Figure 9 nanomaterials-09-01421-f009:**
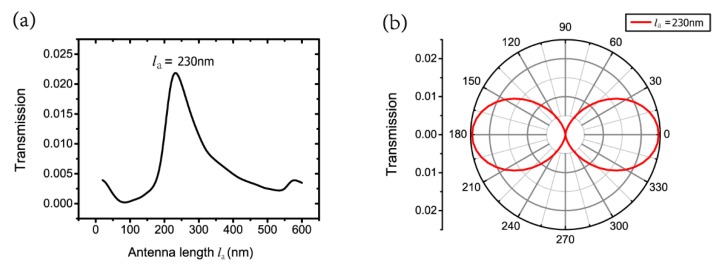
(**a**) Transmission with various antenna lengths of $l_{a}$ under λ=500 nm for TM polarization; (**b**) transmission with different polarization angles at $l_{a}=230\;\mathrm{nm}$ under λ=500 nm.

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
