# Peer review of "Near-Field Enhancement and Polarization Selection of a Nano-System for He-Ne Laser Application"

_nanomaterials, 2019, doi:10.3390/nano9101421_

Round 1
Reviewer 1 Report
The study is of interest and potentially of high significance, and yet a few words concerning the technical practical feasibility of the proposed nano-polarizer at the end (in the conclusions perhaps) would be welcome. Otherwise it remains a nice exercise, a numerical simulation whose implications remain unclear.
Reviewer 2 Report
The manuscript investigates the polarization-dependent transmission through a plasmonic slit, proposing the device as an ultra-compact polarizer. Although the analysis seems solid, I do not see enough novelty and importance to guarantee publication in a high-impact journal such as Nanomaterials. In particular:
[1] The physics of light transmission through plasmonic slits is well-known and thoroughly studied; the proposed structure does not add significant insight nor does it investigate novel phenomena.
[2] In my opinion, the way the proposed device is presented, i.e. as a small-footprint nanopolarizer is rather misleading. It is true that the device has a very small footprint, as dictated by the dimensions of the slit and the additional Ag stripe on-top. However, light is coupled from free-space as a planewave, therefore the total lateral dimensions reported (1200x1200 nm^2) are arbitrarily chosen and do not correspond to the actual functionality. In fact, even for a diffraction-limited focused beam at the investigated wavelengths (a case is not studied in the paper), only a very small portion of the TM polarized light would pass through making the device unpractical; this is indirectly acknowledged in the manuscript by reporting only the normalized transmittance, which is not relevant for a true application. In other words, the extinction ratio of the device might be very high, but for free-space coupling it cannot act as an efficient polarizer.
[3] A free-space nanoscale polarizer can be designed and fabricated using a standard metallic wire grating fabricated e.g. via nanolithography. This would provide true polarizing and broadband behaviour for impinging planewaves, without the complicacy, low transmittance and wavelength-dependent response of the proposed device.
Reviewer 3 Report
It is a good incremental work and good enough to be published as it is.
Author Response
Dear reviewer,
We greatly appreciate your comments on our manuscript. We also have improved our English throughout the paper.
Best wishes.
Round 2
Reviewer 2 Report
I would like to thank the authors for providing a response to my comments. In their reply, the authors stress as the advantage of the proposed polarizer its very small footprint, contrary to nanowire grid polarizers, which represent a standard solution.
Still, I believe that this aspect has not been clarified. The polarizer works for a free-space coupled planewave beam. Therefore, the lateral footprint of the device cannot be univocally determined. In the FDTD calculations the authors select the lateral dimensions of the computational space to be 1200x1200 nm^2. This is arbitrarily chosen. For instance, the absolute transmittance of the device for a Gaussian-beam with 5-micron diameter would be negligible, while for 1-micron larger, but still very low (in both the original and the revised manuscript the authors choose not to report absolute numbers for the transmittance of the pass-polarization). Even for a diffraction-limited focused impinging light beam, the beam dimensions will be significantly larger than the slit aperture, hence very low transmittance is expected.
On the contrary, a wire-grid polarizer’s dimensions can indeed extend to microns or even millimeters, but this is in order to accommodate the size of the impinging beam, not because such sizes are needed for the polarizer to function. In other words, a wire-grid polarizer of the reported footprint of 1200x1200 nm^2 would provide much better polarizing performance and easier manufacturing for a beam spot of that size.
Author Response
Dear reviewer,
Thanks very much for your second-round comments. We agree your opinion that the periodic nanowire grating is a more efficient polarizer.
According to your comments, we have re-examined the advantages of transmission with periodic apertures and with a single aperture. We agree that the transmission with periodic apertures is much higher due to the presence of the period. However, the transmission with a single aperture is also an important issue as it focus on the transmission enhancement of optical near-field and related works (references [21]-[41] in the manuscript) in this area have been cited in the introduction in the new version. Our study focus on the transmission enhancement and polarization selection of the proposed nanosystem in the optical near-field.
According to your comment, we have made corresponding revisions:
We have re-written the title, abstract, introduction and conclusion to make our study explicit. We have presented our results with absolute transmission as your opinion that the normalized-to-area transmission is not intuitive. All the figures related to transmission are re-drawn and absolute transmission values are re-given out. We have also added the figures and descriptions about the near-field spot at the aperture export in Fig. 2 and Fig. 3 for TM and TE polarizations.
Please see the detailed revisions in the new version.
Thank you again for your comments. Your comments do help us improve our manuscript a lot.